# In Vivo Validation of Alternative FDXR Transcripts in Human Blood in Response to Ionizing Radiation

**DOI:** 10.3390/ijms21217851

**Published:** 2020-10-23

**Authors:** Lourdes Cruz-Garcia, Grainne O’Brien, Botond Sipos, Simon Mayes, Aleš Tichý, Igor Sirák, Marie Davídková, Markéta Marková, Daniel J. Turner, Christophe Badie

**Affiliations:** 1Cancer Mechanisms and Biomarkers Group, Radiation Effects Department, Centre for Radiation, Chemical & Environmental Hazards, Public Health England, Chilton, Oxfordshire OX11 0RQ, UK; lourdes.cruzgarcia@phe.gov.uk (L.C.-G.); Grainne.OBrien@phe.gov.uk (G.O.); 2Oxford Nanopore Technologies, Gosling Building, Edmund Halley Way, Oxford OX4 4DQ, UK; Botond.Sipos@nanoporetech.com (B.S.); Simon.Mayes@nanoporetech.com (S.M.); Daniel.Turner@nanoporetech.com (D.J.T.); 3Department of Radiobiology, Faculty of Military Health Sciences in Hradec Králové, University of Defence in Brno, 500 01 Hradec Králové, Czech Republic; ales.tichy@unob.cz; 4Biomedical Research Centre, Hradec Králové University Hospital, 500 01 Hradec Králové, Czech Republic; 5Department of Oncology and Radiotherapy and 4th Department of Internal Medicine—Hematology, University Hospital, 500 05 Hradec Králové, Czech Republic; igor.sirak@fnhk.cz; 6Department of Radiation Dosimetry, Nuclear Physics Institute of the Czech Academy of Sciences, 180 00 Prague 8, Czech Republic; davidkova@ujf.cas.cz; 7Institute of Hematology and Blood Transfusion, 128 00 Praha 2, Czech Republic; marketa.markova@uhkt.cz

**Keywords:** nanopore sequencing, biodosimetry, FDXR, alternative transcript, splice variants, ionizing radiation, qPCR, gene expression

## Abstract

Following cell stress such as ionising radiation (IR) exposure, multiple cellular pathways are activated. We recently demonstrated that ferredoxin reductase (FDXR) has a remarkable IR-induced transcriptional responsiveness in blood. Here, we provided a first comprehensive FDXR variant profile following DNA damage. First, specific quantitative real-time polymerase chain reaction (qPCR) primers were designed to establish dose-responses for eight curated FDXR variants, all up-regulated after IR in a dose-dependent manner. The potential role of gender on the expression of these variants was tested, and neither the variants response to IR nor the background level of expression was profoundly affected; moreover, in vitro induction of inflammation temporarily counteracted IR response early after exposure. Importantly, transcriptional up-regulation of these variants was further confirmed in vivo in blood of radiotherapy patients. Full-length nanopore sequencing was performed to identify other FDXR variants and revealed the high responsiveness of FDXR-201 and FDXR-208. Moreover, FDXR-218 and FDXR-219 showed no detectable endogenous expression, but a clear detection after IR. Overall, we characterised 14 FDXR transcript variants and identified for the first time their response to DNA damage in vivo. Future studies are required to unravel the function of these splicing variants, but they already represent a new class of radiation exposure biomarkers.

## 1. Introduction

Messenger RNAs (mRNAs) are the molecules that translate genomic DNA information into proteins. However, the correlation between mRNA levels and protein abundance varies due to post-transcriptional regulatory mechanisms which play a key role in gene expression. Following transcription, alternative mRNA transcripts can be generated from a single gene, resulting in the translation of different isoforms of the same protein with distinct functions or subcellular localisation. This is an essential mechanism playing an important role in cellular differentiation and organism development [1]. mRNA splicing greatly increase cellular and organismal complexity, providing diversification of gene function. This process is tightly regulated and needs to be executed in time and space so as not to interfere with the normal cellular and organismal physiology. Alternative splicing includes constitutive splicing, intron retention, exon skipping, alternative splice site selection, mutually exclusive splicing, alternative promoter selection, and alternative polyadenylation sites [2]. These alternative transcripts can also arise from the use of alternative transcription start (TSS) and stop sites which has been reported to contribute to transcript diversity even more than alternative splicing [3]. Following DNA damage, alternative transcripts have been detected in human lymphoblastoid cell lines and primary fibroblasts [4]. More specifically, many genes showed alternate transcripts levels as a response to ionizing radiation (IR) exposure, which can also induce the use of secondary promoters to produce alternative transcripts in some genes [5,6]. This has been reported for several DNA repair genes [7].

Gene expression analyses of white blood cells have proven to be an accurate and fast method to assess IR exposure in blood samples for biodosimetry purposes [8,9,10,11,12,13,14]. From arrays and sequencing efforts [14,15,16], a broad number of genes have been identified whose response to IR can qualify as reliable biomarkers of exposure. Nevertheless, further validation is still required, as gene expression-based dose assessment often relies on pre-exposed samples as controls to take into account inter-individual variability in endogenous levels for accurate dose-estimations.

Recently, the gene FDXR has demonstrated its remarkable response to radiation in vivo, giving precise dose estimates in blood samples from total and even partially irradiated radiotherapy patients [8]. FDXR encodes for a protein involved in electron transport and implicated in the biosynthesis of iron–sulphur clusters and in heme formation [17]. Alternative transcripts have been identified for FDXR in vitro and, as IR induces alternative splicing, with some FDXR variants responding differently to IR exposure [18].

Some confounding factors have proven to affect radiation responsive genes such as simulated bacterial infection by lipopolysaccharide (LPS), curcumin, cancer condition, or gender [8,9,19,20]. FDXR response to radiation has been observed to be temporarily counteracted by LPS [8] probably due to the downregulatory effect of LPS on P53 expression [21]. P53 is one of the main transcriptional factors regulating the DNA damage response and its target genes, and their variants could potentially also be affected by LPS.

The emerging long read sequencing technologies have provided a tool for reliable variant calling. Currently, either nanopore sequencing developed by Oxford Nanopore Technologies (Oxford, UK) [22,23,24] and single-molecule real-time (SMRT) sequencing, developed by Pacific BioSciences (PacBio) [25] provide full-length transcripts, which eliminates the issues of next-generation sequencing techniques related with mapping ambiguity [26,27]. Nanopore sequencing has been used in the present study and consists of sequencing a single-stranded DNA/RNA molecule by recording characteristic current changes when the nucleobases pass through a biological pore.

By combining transcript variant-specific quantitative real-time polymerase chain reaction (qPCR) analysis together with long sequencing reads, the present study provides an exhaustive characterisation of ferredoxin reductase (FDXR) alternative transcripts in response to ionising radiation (IR) in human blood ex vivo and in vivo.

## 2. Results

### 2.1. FDXR Expression Profile in Blood Irradiated Ex Vivo

The results showed that all variants present a clear dose-dependent response with large variability in the amplitude of response between them (Figure 1, Appendix A). Variants FDXR-206 (39-fold up-regulation after a dose of 4 Gy) and FDXR-214 (22 fold up-regulation) were the most responsive variants to radiation (Figure 1A) and FDXR-203 the least responsive (seven-fold up-regulation after a dose of 4 Gy) (Figure 1A). When assessing the dose-response correlation (Appendix A), all the variants showed good R^2^ values between 0.85 and 0.93 and Pearson’s correlations between 0.93 and 0.96. For all variants, the response tends to flatten out as the dose increases above 2 Gy.

At the basal level of expression, FDXR-202 showed the highest background level apart from FDXR-201, followed by FDXR-205 (Figure 1B). Interestingly, variant FDXR-217 and FDXR-203 were expressed at very low levels compared to the other variants (Figure 1B). They were almost undetectable, reaching the limit of detection even after 45 cycles of qPCR.

When comparing gender effect on the radiation-response of the variants, no differences were identified apart from variant FDXR-206 at 4 Gy (Figure 2, Appendix A). Regression analysis for each variant for women and men (Appendix A) showed that R^2^ values are slightly lower for women than men for most of the variants. Pearson correlation analyses showed a linear correlation between dose and gene expression for all the variants in both genders except for FDXR-203 in women (Appendix A). The variant presenting the linear correlation with the highest R vale and lower p-vale for women and for men was, respectively, FDXR-217 (R = 0.93, *p*-value 0.002) and FDXR-203 (R = 0.99, *p*-value < 0.001) apart from FDXR-201+.

Comparisons of the radiation response of the different variants in women indicated that although variant FDXR-206 and FDXR-214 show higher fold changes, there are no significant differences with the rest of the variants except for FDXR-203 (Figure 2A). However, in men, variant FDXR-206 was significantly most responsive than the other variants for doses between 1–4 Gy (Figure 2B).

When comparing the basal level of expression of the variants between gender, there were no differences observed (Appendix A).

### 2.2. Effect of Bacterial Endotoxin Exposure on the FDXR Variants Response to Radiation

To address the question of the specificity of the transcription of these transcript variants to IR or at least a clastogen, blood from five male donors was exposed to the bacterial endotoxin LPS (500 ng/mL) 1 h before irradiating the samples at 2 Gy (dose rate 0.5 Gy/min) as a pro-inflammatory stimulus to mimic bacterial infection. The transcriptional response of the variants was analysed either after 2 h or 24 h post-exposure. The results demonstrated differences in the response to LPS versus radiation after LPS treatment only in some variants (Figure 3). After LPS treatment, at the 2 h time point, the gene expression levels of variant FDXR-204, FDXR-202, and FDXR-205 were slightly downregulated. When LPS was combined with IR, variants FDXR-201, 214, 204, 202, 217, 205, and 206 showed a decrease in their response to radiation after 2 h post-exposure. However, this counteractive effect of LPS was mostly dissipated at the 24 h time point except for variant FDXR-201 and 202.

Besides, looking at the early transcriptional response, all the variants were significantly upregulated after 2 h after 2 Gy exposure apart from FDXR-203 and FDXR-217.

### 2.3. Expression Profile of FDXR Variants after Irradiation In Vivo

The expression profiles of the different FDXR variants were then analysed in vivo in blood samples of TBI patients before and after irradiation (Figure 4). The basal levels of expression of the different variants showed variability between patients possibly reflecting the condition of these leukaemia patients (Figure 4A). The variants presenting a highest trend of expression at basal level were FDXR-202, 205, and 204 although this is mainly driven by one patient, and the lowest was FDXR-217 and FDXR-203, as previously observed ex vivo, even though there were no significant differences between them. After IR exposure, overall, the highest fold of change in expression was 10.1-fold for FDXR-204 (see Appendix A), although the up-regulation was not found to be significant due to the large inter-individual variability in response in these patients. The most responsive variants to radiation were variant FDXR-206 for patient TBI1 and TBI4, FDXR-204 TBI3 while TBI2 hardly showed any up-regulation apart from FDXR-214, although it is weak (Figure 4B). The only variant which presented a consistent and significant upregulation was variant 214 (Figure 4B and Appendix A).

We also analysed the variant usage in radiotherapy patients where the dose received is localised to the tumour area, and the dose received in each fraction represents a much lower dose to the blood i.e., circulating white blood cells (Figure 5). Endometrial cancer patients blood samples were collected 24 h after the first radiotherapy fraction; importantly, all the variants presented a significant up-regulation as compared to their respective control sampled prior to the first radiotherapy fraction was delivered (Figure 5B). The average fold-change was consistent between variants (between 3.1-fold and 2.4-fold-change), with variant FDXR-206 presenting the highest upregulation (3.1-fold) followed by variant FDXR-204 (2.9 fold) (Figure 5B), although no significant differences were found between variants. The variant FDXR-205 response was the most consistent between the endometrial patients (Appendix A) with a coefficient of variation (CV) of 19.6. The least consistent variants were FDXR-214 and FDXR-217 with a CV of 46 for both variants (Appendix A).

At basal level, variant FDXR-202 was the second highest expressed after FDXR-201+ mRNAs, whose primers include all the other variants (Appendix A and Figure 5A).

### 2.4. Sequencing Analyses

In order to obtain enough polyA+ mRNA, long read nanopore sequencing data were generated using PBMCs from three healthy donors per replicate (total of nine donors for three experiments), either sham or 2 Gy exposed. Sequencing analysis identified 12 FDXR variants (Appendix A), six of them FDXR-207, FDXR-208, FDXR-209, FDXR-213, FDXR-218, and FDXR-219 not previously characterised by qPCR. Amongst them, seven variants presented a significant response to IR, FDXR-201 as expected and FDXR-204, FDXR-206, FDXR-208, FDXR-213, FDXR-218, and FDXR-219 (Figure 6). Amongst the seven, FDXR-206, FDXR-218, and FDXR-219 were very low expressed while FDXR-218 and FDXR-219 remarkably showed an undetectable endogenous levels (no counts presented in the control sample), even after generating over 50 million reads of mRNA per sample [28] (Figure 6).

## 3. Discussion

In the present study, we characterised the differential isoform expression of FDXR following IR exposure in human blood ex vivo as well as in vivo with variant-specific qPCR primers and full-length transcripts obtained from nanopore long sequencing reads; we have identified differential expression isoforms and provided the first comprehensive analysis of the transcriptional complexities of FDXR variants. The identification of FDXR as a gene responsive to IR in lymphoblastoid cell lines was first described by Rieger and Chu (2004) [29]. A few years later, its transcriptional upregulation to IR was also detected in human primary fibroblast [30] and blood samples [16]. FDXR potential as a biomarker of radiation exposure has been continuously studied mainly in human blood samples exposed ex vivo [11,12,13], until recently, where FDXR expression profiles were used to calculate dose estimates in blood samples exposed in vivo from radiotherapy patients [8,31]. From high (TBI) to very low levels of exposure (CT scans), accurate dose estimation was possible, and this extended to locally irradiated patients irrespective of the radiotherapy treatment and cancer type.

Alternative splicing has a role in almost every aspect of protein function and has a central role in gene expression [32]. We, therefore, decided to investigate the splicing activity for FDXR in the context of DNA damage induced by IR exposure in vitro and its confirmation in vivo in humans. The FDXR gene has a repertoire of 20 alternative transcripts (https://www.ensembl.org/Homo_sapiens/Gene/Summary?g=ENSG00000161513;r=17:74862497-74873031) and its alternative splicing pattern is altered in response to IR [4,18]. However, first, it was still unknown which variants are affected; second, if long read sequencing would permit the identification of different variants; and last, crucially, if it could be confirmed in humans in vivo.

Our results showed that twelve FDXR variants, from the fourteen identified and measured in blood and PBMCs, responded significantly to IR. Their response showed slightly different degrees of responsiveness and time required for their transcriptional upregulation post-exposure. FDXR-201 is by far the most endogenously expressed variant and FDXR-206 was the most responsive to IR reaching a 40-fold increase in expression following exposure to the highest dose we have used 4 Gy. As the dose response is clearly flattening for the highest doses we used, it would not be anticipated that the fold of change would increase drastically in response to an exposure to higher doses.

We addressed the role of gender by comparing males and females’ donors. FDXR-206 was the most responsive to IR in men. Sex differences in gene expression have been widely reported [33], and differences in gene variants distribution between genders have also been identified in humans [34,35] and mice [36]. Even though the gender groups were relatively small (*n* = 5) in the present study, gender bias was examined in eight variants, but no significant differences were identified apart from FDXR-206 at the single highest dose of 4 Gy. Correlation analyses reinforced this lack of gender-dependent differences, apart from variant FDXR-203. In this case, the low level of expression measured by qPCR and sequencing might explain why its linear dose-response is affected in women specifically.

Previous research on FDXR gene expression response to IR have shown that its response is higher at 24 h than 2 h post exposure [12], although already presenting a significant upregulation at 2 h [8]. These results are comparable to the present FDXR-201+ profiles, since the same primers were used in those studies. This response pattern is similar in most of the variants identified by qPCR except for FDXR-203 and FDXR-217 which didn’t show an upregulation at 2 h post-exposure. Although these two variants are clearly responsive to radiation, their endogenous level of expression is low and FDXR-217 was not even detected by sequencing. One explanation is that the qPCR primers for FDXR-217 were covering FDXR-213 and FDXR-218 variants, hence the up-regulation detected by qPCR could be mainly due to the contribution of the detection in the FRXR-217 dose-response profile. Therefore, we conclude that low levels of expression could be affecting their detection at early time points and their dose–response relationship.

Nanopore sequencing data showed that FDXR-204, 208, 206, and 201 show the strongest response to radiation (fold changes of 28, 21, 20, and 19, respectively), hence the contribution of variant FDXR-208 in the FDXR-206 group is important in the overall FDXR-206/208/209 response. From the 12 variants identified by sequencing, eight of them were very lowly expressed, with only three of them showing a significant upregulation after exposure (FDXR-206, FDXR-218, FDXR-219). At background level, the variants presented differences in expression and interestingly, variants FDXR-218 and FDXR-219 did not present counts in the control samples, even at the very high level of sequencing of polyA+ mRNA only (none in 50 million long reads on average per sample). This lack of expression could be very useful in the field of biodosimetry to assess radiation exposure in case of a nuclear accident but also important for assessing exposure to more common sources of IR such as in medical and occupational settings. Gene-expression analyses provides fast and accurate dose estimates, with FDXR recently demonstrated to be an outstanding biomarker ex vivo as well as in vivo [8]. Nevertheless, it is detectable in non-exposed individuals preventing its use for triage purposes of exposed versus non-exposed. Future work will now need to determine if the expression of FDXR-218 and FDXR-219 is specific to IR exposure and presents clear dose-dependent relationships. Given their specific expression patterns, these two variants represent a new exciting development in the field of biomarkers of exposure.

Alternative transcripts are a source of protein diversity which can lead to loss of protein function, transcript instability, faulty localization of a protein, and different or opposite functions [32,37,38]. However, alternative gene transcripts can also lead to non-coding mRNAs or mRNA containing premature termination codons (PTC) [39,40]. From the different FDXR variants identified in this study, three of them (FDXR-206, FDXR-209, FDXR-218) have a PTC which cause the degradation of the mRNA by non-sense mediated decay pathway. This pathway prevents the generation of truncated proteins which can lead to a deleterious effect in the organism [41]. Looking at the sequencing profile, these three variants are very low expressed, even though FDXR-206 and FDXR-218 show significant response to IR.

Two of the variants identified by sequencing, FDXR-207 and FDXR-208 have retained an intron in their sequence and only FDXR-208, a non-coding transcript, shows a high background level of expression together with a strong response to IR. Non-coding transcripts from a coding gene have been proposed to have regulatory effects on their own genes by acting as competitors of the mRNAs for miRNA silencing [39,42]. Therefore, the high response of FDXR-208 in response to DNA damage could potentially be involved in this miRNA sponge function for variant FDXR-201. This role of FDXR-208 would allow higher translation of FDXR-201, hence providing higher amount of FDXR protein needed by the cell in response to IR. However, further functional studies are required to prove this hypothesis.

Aside from FDXR-208, sequencing analysis showed a strong up-regulation of variants FDXR-213 and FDXR-204 following IR exposure. These two variants codify for two proteins of drastically different size (179 and 534 amino acids, respectively), suggesting different functionality of these proteins. Although proteomic analysis reported so far showed low levels of alternative isoforms [43], suggesting a role of variants at mRNA level, the variants FDXR-213 and FDXR-204 have been identified at protein level (UniProtKB accession numbers J3QQW7 and A0A0A0MT64) [44].

Although variant FDXR-205 doesn’t seem to stand out for having a particular response to radiation, it requested specific attention as this variant has an alternative TSS. Multiple TSSs are common in human genes allowing the production of different transcripts from the same gene [45,46]. Adaptive purposes have been attributed to this phenomenon [47], by generating different proteins from the same gene with even opposite functions [48]. However, alternative transcription initiation has also been proposed as nonadaptive, suggesting that there is only one optimal TSS per gene and the other transcripts produced from different TSS are errors from transcriptional initiation [49]. IR-induced alternative transcription products generated from an alternative TSS have been previously identified for several genes involved in DNA repair such as the ribonucleotide reductase regulatory TP53 inducible subunit M2B (RRM2B) and the XPC complex subunit, DNA damage recognition and repair factor (XPC) [7]. In the case of the FDXR-205 variant, it is transcribed through an alternative TSS which creates a transcript differing in the 5′ untranslated region (Appendix A). Although our results cannot contribute in clarifying these two hypotheses, in vivo and ex vivo experiments showed the transcript is not only expressed at background level but also showing a good dose-response correlation in the range 0.25–4 Gy. However, sequencing analysis showed a very low frequency for this variant. One hypothesis is that this difference is due to a difference in the protocol used; qPCR analysis was performed with whole blood containing a large proportion of granulocytes while PBMCs were isolated for sequencing analysis.

We previously addressed the role of several potential confounding factors on gene expression after DNA damage [9], such as anti-inflammatory/anti-oxidant agents and simulated bacterial infection, showing that it can affect the transcriptional response to IR of some genes [8,9,19]. We showed that infection simulated by administration of LPS ex vivo has a mild effect counteracting FDXR response to IR shortly after LPS stimulation [8]. We, therefore, wanted to assess the role of LPS in individual FDXR variants; similar effects were observed in variant FDXR-214, 204, 202, 217, 205 and 206, where LPS counteracted their response to IR after 3 h of LPS administration and 2 h post-radiation exposure. This effect is practically lost after 24 h, apart from variant FDXR-202. LPS is known to downregulate p53 [21], a key player in the DNA damage response [50] and the transcription factor driving FDXR transcription [51]. Therefore, the regulatory mechanism of LPS observed here probably occurs through p53 and it affects splicing and FDXR variants.

To the best of our knowledge, transcript variant response to IR exposure in radiotherapy patients has not been addressed before. To this end, we first performed in vivo analyses of FDXR variants in TBI patients; although doses were theoretically homogeneously absorbed in TBI patients, our data showed patient-dependent background levels and responses to IR in terms of expression of the variants. TBI patients were first diagnosed with ALL, which means the presence of a malignant transformation of lymphoid progenitor cells in the bone marrow and an increased proliferation [52]. Blood obtained from these patients contains high numbers of immature lymphoblasts which changes the overall blood sample expression profile [53] and potentially the response to IR compared to a non-treated blood sample. Different FDXR background expression levels have been described in TBI patients before [8]. Besides, aberrant splicing has been described in myelodysplastic syndromes and myeloid leukaemia due to mutations in splicing factors, affecting isoforms ratios [54]. This aberrant splicing mechanism could potentially explain the differences observed in the variants’ expression levels in TBI patients compared to healthy donors.

When looking at the response of the variants to the high dose whole body radiation exposure, the folds-changes observed showed large variability and only one variant, FDXR-214, was significantly different from the controls. Retrospectively, it would be interesting to check the mutation profile of the leukaemic cells for these patients. Our data raise the possibility that different patient pathophysiological stages could explain the rather unexpected differences in FDXR variants basal levels of expression and their response to IR.

Studying FDXR transcript variants in endometrial cancer patients was theoretically more challenging as the dose of radiation is only delivered locally with a total dose to the entire blood volume being very low [8,55]. Surprisingly, relatively similar expression profiles between variants were found with responses to IR, ranging between 2.4-fold and 3.1-fold changes, and presenting some degree of variability between patients. FDXR-205 showed the most consistent response between patients. Despite the rather limited number of patients it could be noted that variants FDXR-214, FDXR-217, and FDXR-203 presented higher inter-individual variability. IR-induced expression of certain gene variants or IR-induced modulation of alternative splicing have been previously associated with radioresistance [56,57] and radiosensitivity [58]. These are key factors involved in radiotherapy failure, leading to adverse effects such as toxicity or cancer recurrence. Although we do not have relevant clinical information on the patients who provided blood samples for this study, it would certainly be interesting in future studies to assess the role of certain variant response to an adverse outcome. From our data, it could be suggested that future attention should be directed to variants FDXR-214, FDXR-217, and FDXR-203 due to the strong variability in their response between patients and to determine their prognostic value for treatment outcome, normal tissue radio-sensitivity, and long-term adverse effects such as second cancer induction.

Analysis of the background level of expression in endometrial cancer patients indicated very low levels of expression of some variants such as FDXR-217 and FDXR-203, confirming the previous observations from ex vivo samples of healthy donors. However, the expression level of two variants, FDXR-202 and FDXR-205, compared to the rest of the variants is lower in endometrial cancer patients than in healthy donors. Changes in transcript usage have been observed for several genes in cancer patients and this represents a potential application as cancer biomarkers [59,60]. Further analysis of a larger cohort of patients and healthy donors would be required to identify a role of FDXR-202 and/or FDXR-205 as potential cancer biomarkers.

This study provides novel insights into FDXR alternative transcripts as a mechanism for response to DNA damage and cell stress. Ex vivo analysis of FDXR in blood and PBMCs showed responsiveness to radiation for most of the variants. Gender and inflammation seem not to exert a major confounding effect in the extent of the splicing observed. In vivo analysis showed inter-individual variability in variants frequency between radiotherapy patients and differences in endogenous levels which possibly reflects the patients’ pathophysiological state. Nanopore sequencing permits long reads which allowed to identify seven variants which could not be detected individually by qPCR. Although this is a comprehensive study of the FDXR variants in human blood at transcriptional level, future functional studies are required to understand their biological role.

It would be of great interest to study in more detail the activation of the spliceosome following IR. Some specific FDXR transcript variants are hardly (or not at all) detectable in PBMCs from control blood samples either in vitro or in vivo in patients, raising the possibility that their activation has very specific purposes with important consequences at the cellular and tissue level. The transcriptional response to cellular stress is critical for cell survival and novel insights about how FDXR alternative splicing regulation contributes to the progression of a variety of diseases would be of particular interest. Importantly, during radiotherapy cancer treatment, a percentage of patients experience severe normal tissue toxicity reactions [61,62] and identification of those patients at the beginning of the treatment, possibly via measuring radiation-induced alternative splicing, would be essential for personalized therapy with improved prognosis.

## 4. Material and Methods

### 4.1. Blood Collection and Irradiation Ex Vivo

Peripheral blood samples freshly collected from 10 healthy donors (five men and five women; age range: 35–60 years) were collected and exposed to a range of X-ray doses (500 µL of blood; 0.25, 0.5, 1, 2, 3, and 4 Gy at a dose rate 0.5 Gy/min). An HS X-ray system (AGO X-Ray Ltd., Aldermaston, UK) (output 13 mA, 250 KV peak) was used to irradiate the samples. Blood samples were kept at 37 °C in an incubator with 5% CO_2_ for 24 h after exposure. After 24 h, the blood was mixed with 1 mL of RNA later (Thermo Fisher Scientific, Loughborough, UK) and stored at −80 °C until being processed for RNA extraction.

For the experiment on lipopolysaccharide (LPS) effect, blood samples from five healthy donors (five men, age range: 35–60 years) were incubated with LPS at 500 ng ml^−1^ (stock at 1 mg/mL in 50% ethanol, Sigma-Aldrich, Irvine, UK). LPS was added to 500 µL of blood 1 h before being either mock-irradiated or exposed to a 2 Gy X-ray dose (dose rate 0.5 Gy/min). Blood samples were kept at 37 °C in an incubator with 5% CO_2_ for 2 h and 24 h after exposure. After the incubation time, the blood was mixed with 1 mL of RNA later and stored at –80 °C. Venous blood was taken at the Centre for Radiation, Chemical, and Environmental Hazards Public Health England (Chilton, UK) with informed consent and the ethical approval of the West Midlands–Solihull Research Ethics Committee (REC 14/WM/1182).

### 4.2. Radiotherapy Patient Samples

Blood from total body irradiated (TBI) leukaemic patients and partial body irradiated (PBI) endometrial cancer patients were collected after different times post-exposure. TBI patients, who were diagnosed with acute lymphoblastic leukaemia (ALL), received two-consecutive fractions of radiotherapy, the first one in the morning and the second in the evening with a 12 h interval (total doses received: TBI1 3.2 Gy, TBI2 3 Gy, TBI3 4 Gy, TBI4 4 Gy). Samples from endometrial cancer patients were previously described [55]. Briefly, blood samples were collected into PAXGene tubes (BD Biosciences, Wokingham, Berkshire, UK) before radiotherapy treatment and 24 h after the first fraction (1.8 Gy) after exposure. The collection of blood samples from two TBI (one female and one male, TBI1 and TBI2) and eight PBI endometrial cancer patients was performed at the University Hospital in Hradec Králové (Czech Republic). This study was carried out in accordance with the recommendations of The Code of Ethics of the World Medical Association Declaration of Helsinki (approval no: 201401-S15P) with written informed consent from all subjects. All subjects gave written informed consent in accordance with the Declaration of Helsinki. The protocol was approved by the Ethical Committee of University Hospital in Hradec Kralove (the Czech Republic). Blood samples from the other two TBI patients (two males, TBI 3 and 4) were obtained from Hospital Na Bulovce, Prague, the Czech Republic. The local “Ethics Committee on Trial on Human Medicine Products” approved this study under the code10 February 2017.

### 4.3. RNA Isolation and Reverse Transcription

Total RNA from blood samples exposed ex vivo to X-rays was extracted using a RiboPure™ Blood RNA Purification Kit (Thermo Fisher Scientific, Loughborough, UK). Total RNA from samples collected in PAXgene tubes from radiotherapy patients was extracted with the PAXgene Blood miRNA Kit (QIAGEN, PreAnalytiX GmbH, Hilden, Germany) using a robotic workstation Qiacube (QIAGEN, Skelton House, Lloyd St., N., Manchester M15 6SH, UK). The quantity of isolated RNA was determined by spectrophotometry with a ND-1000 NanoDrop (Thermo Fisher Scientific, Waltham, MA, USA) and quality was assessed using a Tapestation 220 (Agilent Technologies, Santa Clara, CA, USA). cDNA was prepared from 350 ng of the total RNA using a high capacity cDNA reverse transcription kit (Applied Biosystems, Foster City, CA, USA) according to the manufacturer’s protocol.

### 4.4. Quantitative Real-Time Polymerase Chain Reaction

We generated transcriptional dose-response of different FDXR variants in human blood exposed ex vivo to a range of X-ray doses (0.25, 0.5, 1, 2, 3, 4 Gy; dose rate 0.5 Gy/min) by qPCR. Specific primers were designed to identify seven curated FDXR variants (https://www.ncbi.nlm.nih.gov/gene/2232) and an extra pair of primers to detect all the curated variants together with the main expressed variant (FDXR-201, ENST00000293195) (Appendix A). Primers for variants FDXR-204, 217, and 206 cover groups of FDXR variants listed in Appendix A.

Quantitative real-time polymerase chain reaction (qPCR) was performed using a Rotor-Gene Q (QIAGEN, Hilden, Germany) with PerfeCTa SYBR^®^ Green SuperMix (Quanta Biosciences, Inc., Gaithersburg, MD, USA). The samples were run in duplicates in 10 µL reactions with 1 µL of the cDNA synthesis reaction together with primer sets for target FDXR variants (Appendix A) at 500 nM concentration each. The reactions were performed with the following cycling conditions: 2 min at 95 °C, then 45 cycles of 10 s at 95 °C and 60 s at 60 °C. Data were collected and analysed by Rotor-Gene Q Series Software. Gene target Ct (cycle threshold) values were normalized to hypoxanthine phosphoribosyltransferase 1 (HPRT1) internal control (HPRT1 F: 5′ TCAGGCAGTATAATCCAAAGATGGT 3′, R: 5′ AGTCTGGCTTATATCCAACACTTCG 3′). Ct values were converted to transcript quantity using standard curves obtained by serial dilution of PCR-amplified DNA fragments of each gene. The linear dynamic range of the standard curves covering six orders of magnitude (serial dilution from 3.2 × 10^−4^ to 8.2 × 10^−10^) gave PCR efficiencies between 90% and 103% for each gene with R2 > 0.998. Primer specificity was assessed by melting curve analysis (Appendix A).

### 4.5. Blood Irradiation Ex Vivo and RNA Extraction for Sequencing Analysis

Blood from nine healthy donors was collected and exposed to 0 Gy or 2 Gy (10 mL of blood each, dose rate 0.5 Gy/min). After irradiation, the peripheral blood mononuclear cells (PBMCs) were isolated using Histopaque-1077 (Sigma Aldrich, Poole, Dorset, UK) and maintained in LGM-3 culture medium (Lonza, Slough, UK) at 2 × 10^6^ cells ml/mL for 24 h at 37 °C in a humidified 5% CO_2_ atmosphere. After 24 h, the PBMCs were pooled in groups of three donors each, and the RNA was extracted using the RNeasy Midi kit (Qiagen, Manchester, UK). The quantity of isolated RNA was determined by spectrophotometry with a ND-1000 NanoDrop (Thermo Fisher Scientific, Waltham, MA, USA) and quality was assessed using a Tapestation 220 (Agilent Technologies, CA, USA). Venous blood was taken at the Centre for Radiation, Chemical and Environmental Hazards Public Health England (Chilton, UK) with informed consent and the ethical approval of the West Midlands Solihull Research Ethics Committee (REC 14/WM/1182).

### 4.6. Nanopore Sequencing Analysis

The full-length sequencing carried out by Oxford Nanopore Technologies was performed in a PromethION sequencer (Oxford Nanopore Technologies, Oxford, UK) with the libraries prepared using the PCS109 kit according to the instructions (https://community.nanoporetech.com/protocols/cdna-pcr-sequencing_sqk-pcs109/v/PCS_9085_v109_revC_04Feb2019). Before preparing the library, the RNA was poly(A)+ enriched using an Oligotex mRNA Mini Kit (Qiagen, Manchester, UK), and the cDNA was prepared from 1 ng of RNA using strand-switching and VN primers. After the cDNA synthesis, a selective PCR amplification was performed for full-length transcripts before adding specific adapters to start the sequencing run in the PromethION sequencer. Details can be found in Cruz-Garcia et al. (2020) [28].

### 4.7. Nanopore Data Analysis

Nanopore cDNA reads were analysed using a snakemake pipeline modified to handle paired samples [63]. The pipeline (https://github.com/nanoporetech/pipeline-transcriptome-de/tree/paired_dge_dtu) maps the reads to the transcriptome using minimap2 [64] and estimates per-transcript read counts using salmon [65].

### 4.8. Statistical Analysis

Statistical analyses were performed using Minitab software (State College, PA, US). Data are presented as means ± standard deviation (SD) or standard error of the mean (SEM). Comparisons were analysed by an unpaired *t*-test (student’s *t*-test) or a paired t-test. A significance of *p* ≤ 0.05 was applied to all statistical tests performed. Statistical analyses were performed in log transformed data. Multiple comparisons were performed with one-way ANOVA followed by Tukey’s tests. Pearson’s correlation and linear regression analyses were performed to verify dose-response relationships. To measure the consistency of the gene expression response to radiation in the in vivo experiments and sequencing analysis, the coefficient of variation (CV, mean-normalized standard deviation) was calculated for all the variants.

## Figures and Tables

**Figure 1 ijms-21-07851-f001:**
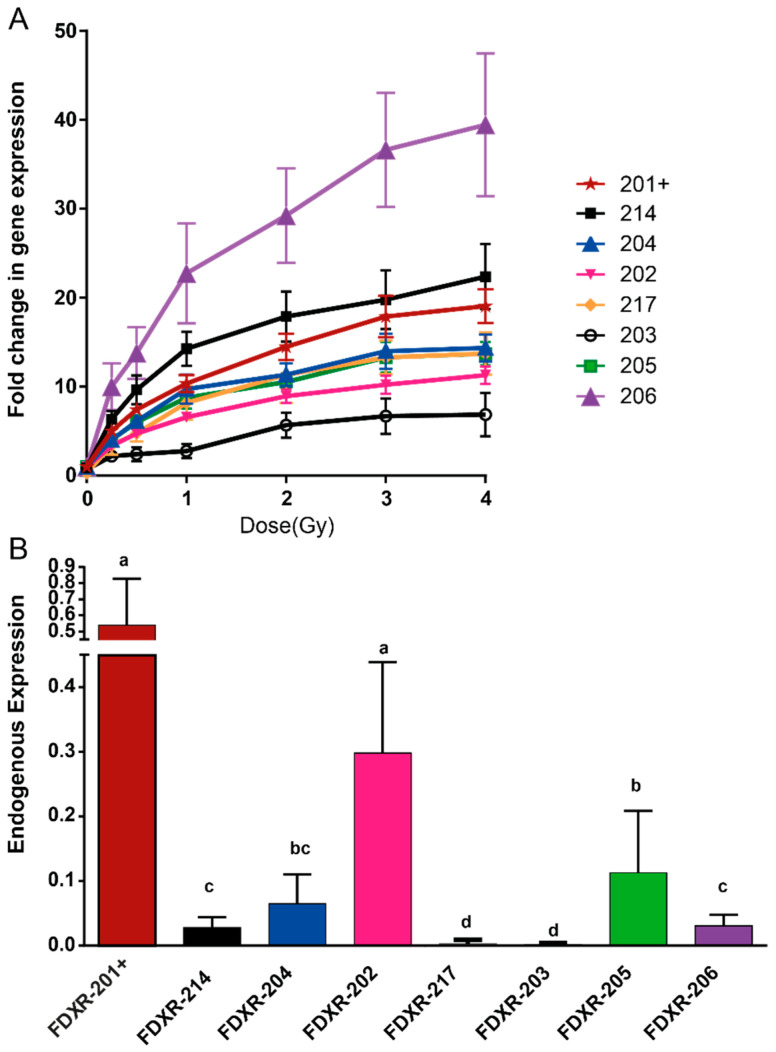
Gene expression of ferredoxin reductase (FDXR) variants in human blood from 10 healthy donors after X-ray irradiation ex vivo (**A**) and at endogenous level (**B**). Blood from five women and five men was exposed to a range of x-ray doses (0.25, 0.5, 1, 2, 3, 4 Gy; dose rate 0.5 Gy/min) and incubated for 24 h at 37 °C. The data are presented as mean values ± standard error of the mean (SEM). One-way ANOVA followed by Tukey’s test was performed for statistical analysis. Values not sharing a common letter are significantly different.

**Figure 2 ijms-21-07851-f002:**
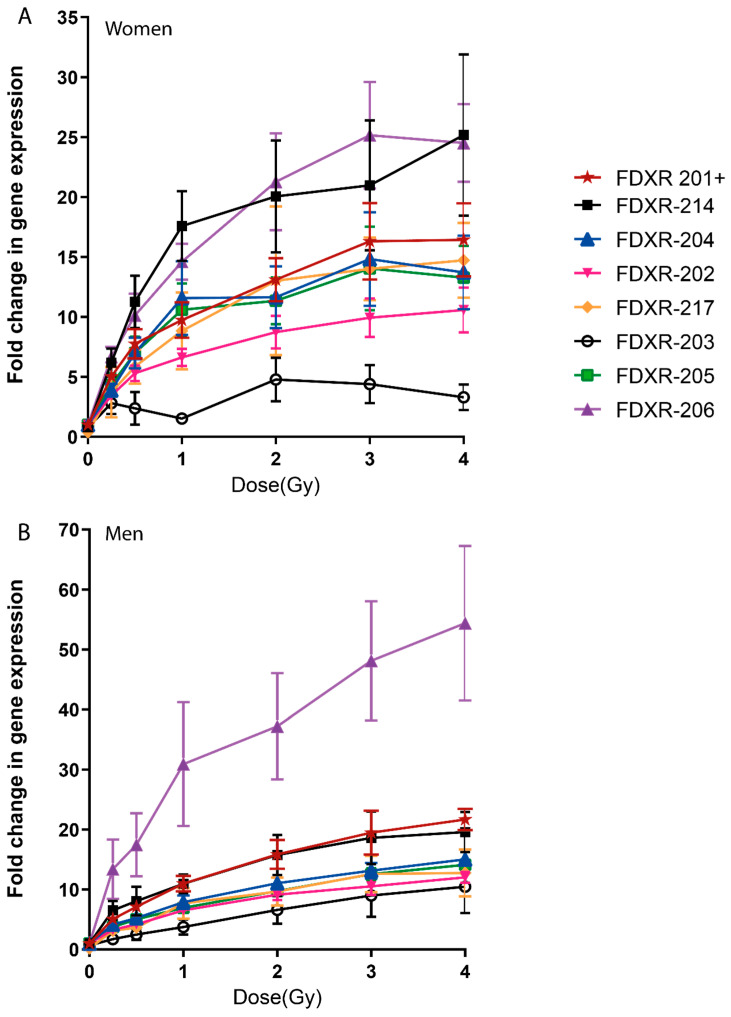
Comparison of gene expression FDXR variants in human blood irradiated ex vivo in women (**A**) and men (**B**). Blood from five women and five men was exposed to a range of X-ray doses (0.25, 0.5, 1, 2, 3, 4 Gy; dose rate 0.5 Gy/min) and incubated for 24 h at 37 °C. The data are presented as mean values ± standard error of the mean (SEM). Statistical analyses were performed in log-transformed data. One-way ANOVA followed by Tukey’s test was performed for statistical analysis.

**Figure 3 ijms-21-07851-f003:**
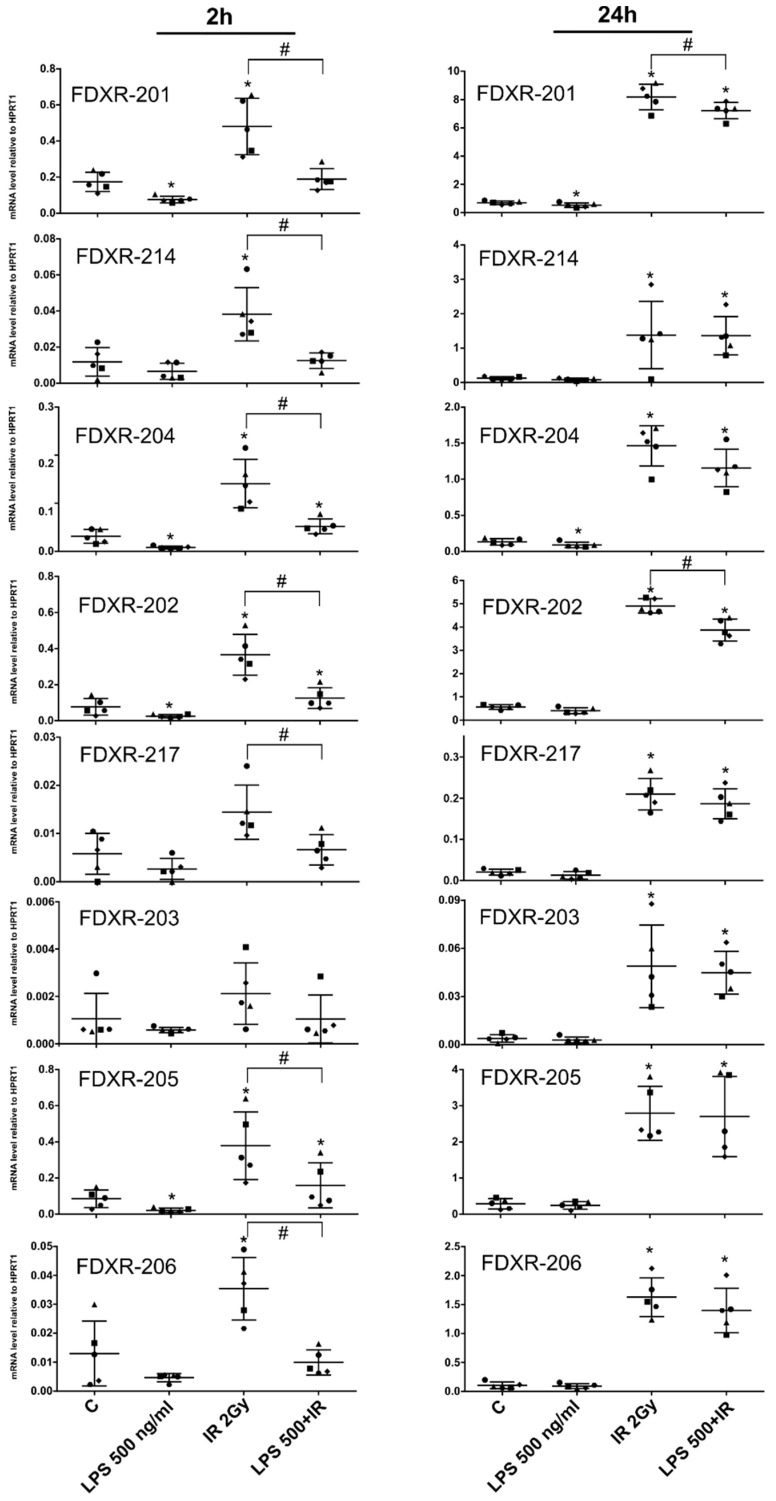
Gene expression profile of FDXR variants in human blood irradiated and/or stimulated with LPS ex vivo. Blood from five male donors was incubated with LPS (500 ng/mL) 1 h before irradiation (2 Gy, 0.5 Gy/min). FDXR variants expression profile was analysed 2 h and 24 h post-irradiation. Data are shown as mean ± SD (*n* = 5). Statistical analyses were performed in log-transformed data. Significant differences (paired *t*-test, *p* ≤ 0.05) with the control were indicated with an asterisk (*) and differences with IR with a hash (#) (only for IR groups).

**Figure 4 ijms-21-07851-f004:**
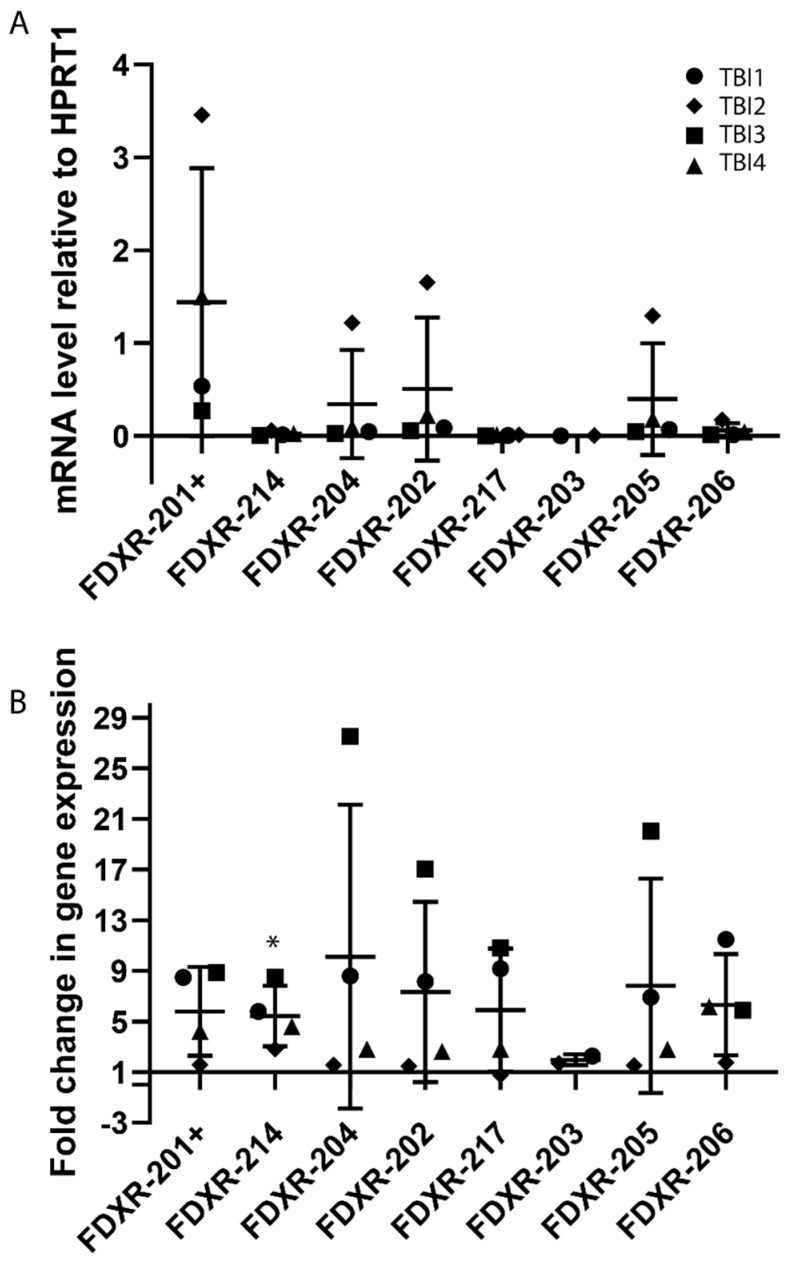
Gene expression profile of FDXR variants in blood samples from total body irradiated patients. The variants basal level of expression was compared between the total body irradiated (TBI) patients (**A**). Total body irradiated patient’s samples (*n* = 4) were collected 24 h after the first fraction and 12 h after the second fraction with total received doses of 3.2 (TBI1), 3 (TBI2), 4 (TBI3), and 4 Gy (TBI4) respectively. The fold change is represented in the graph and values over 1 indicate an increase with respect to control values (**B**). Individual data points are shown for all patients, together with the mean ± SD (each patient is represented with a different symbol). One-way ANOVA followed by Tukey’s test was performed for statistical analysis. No significant differences were found between variants. Significant differences (paired *t*-test, *p* ≤ 0.05) with the control were indicated with an asterisk (*) in panel B.

**Figure 5 ijms-21-07851-f005:**
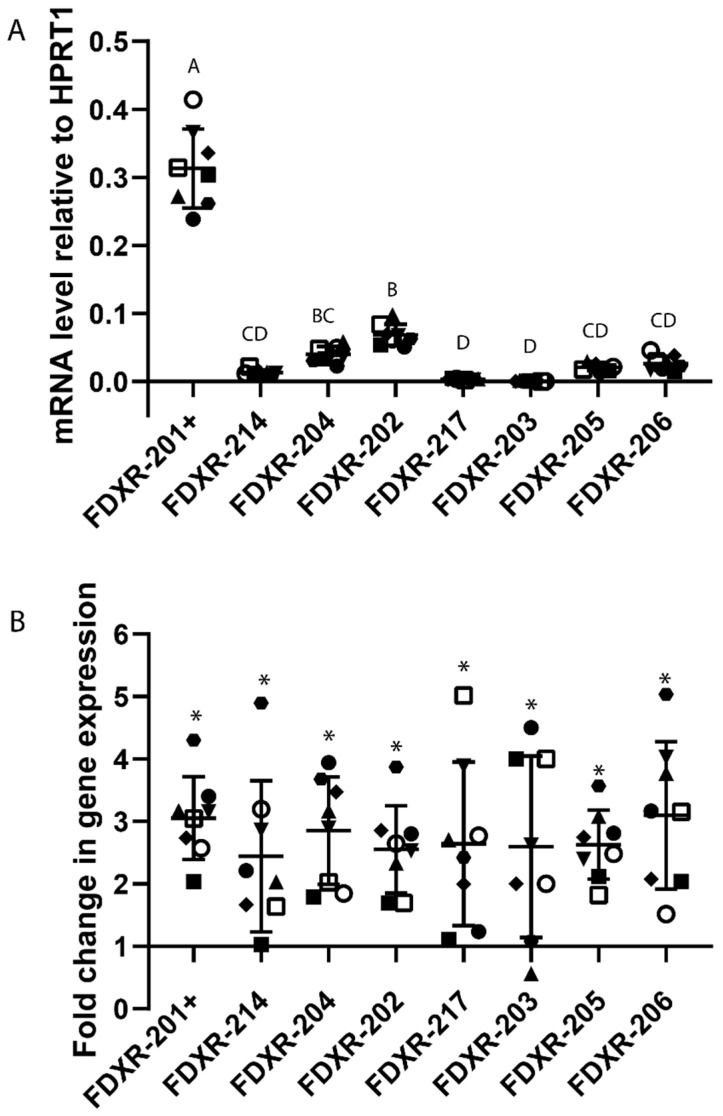
Gene expression profile of FDXR variants in blood from Endometrial cancer patients treated with radiotherapy. The variants basal level of expression was compared between the patients (*n* = 8) (**A**). FDXR variants fold of change in expression 24 h after the first fraction (1.8 Gy) are represented in panel B. The fold change is represented in the graph and values over 1 or below 1 indicate an increase or decrease with respect to control values (**B**). Individual data points are shown for all patients, together with the mean ±SD (each patient is represented with a different symbol). One-way ANOVA followed by Tukey’s test was performed for statistical analysis. Values not sharing a common letter are significantly different. No significant differences were found between variants in panel B. Significant differences (paired *t*-test, *p* ≤ 0.05) with the control were indicated with an asterisk (*) in panel B.

**Figure 6 ijms-21-07851-f006:**
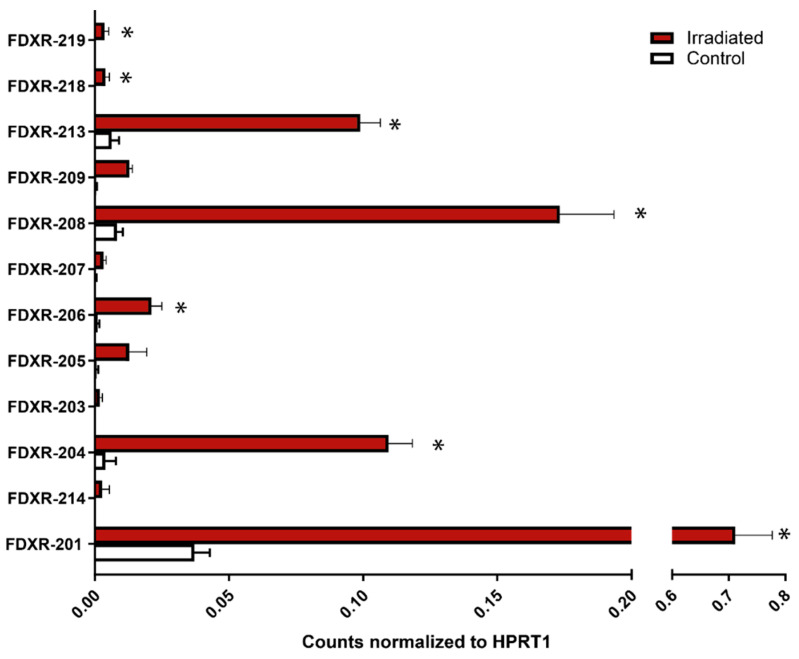
Variants identified by nanopore sequencing. Blood samples were expose to a 2 Gy dose (0.5 Gy/min) and incubated at 37 °C for 24 h. RNA was extracted and poly A+ enriched before preparing the library using a cDNA-PCR kit (PCS109). The sequencing was performed in a PromethION (Oxford Nanopore Technologies). Counts were normalized by HPRT1. Data are shown as mean ± SEM (*n* = 3). Statistical analyses were performed in log-transformed data. Significant differences (paired *t*-test, *p* ≤ 0.05) with the control were indicated with an asterisk (*).

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
