# Peer review of "In Vivo Validation of Alternative FDXR Transcripts in Human Blood in Response to Ionizing Radiation"

_ijms, 2020, doi:10.3390/ijms21217851_

Round 1

Reviewer 1 Report

The presented study touches on one of the most unresearched areas of genomics: the functional significance and regulation of alternative variants of gene splicing. Using variant-specific primers for RT-PCR and new approach with nanopore sequencing, the authors evaluate the background and radioinduced expression levels of the splice variants of the FDXR gene.

The authors took into account all the comments and suggestions that I made when reviewing this work for the first time. I have no more comments on the manuscript and believe that it can be published in its present form.

Author Response

Reviewer 1

The presented study touches on one of the most unresearched areas of genomics: the functional significance and regulation of alternative variants of gene splicing. Using variant-specific primers for RT-PCR and new approach with nanopore sequencing, the authors evaluate the background and radioinduced expression levels of the splice variants of the FDXR gene.

The authors took into account all the comments and suggestions that I made when reviewing this work for the first time. I have no more comments on the manuscript and believe that it can be published in its present form.

We thank again the reviewer for his/her comments in the first revision and we are glad to hear that the manuscript is ready for publications now.

Reviewer 2 Report

Dear authors,

This work presents an original result with significant content about possible biomarkers of radiation exposure. However, the manuscript has some major changes that need to be corrected.

Specific comments:

Point 1: many parts of the text are highlighted in colour. Please remove these marks.

Point 2: Please add line number to facilitate review.

Point 3: The title "Introduction" should be in the same format as the rest of the titles of the work (bold type).

Point 4: Add a citation supporting the phrase "provide full-length transcripts which eliminates the issues of next-generation sequencing techniques related with mapping ambiguity."

Point 5: Authors should include information about the LPS effect in the introduction.

Point 6: The sample size used in the study is low. Is there a bibliography about proven results with similar sample sizes? If not, the authors should include a larger number of samples.

Point 7: The blood samples from the LPS experiment were incubated for 2 or 24 hours? It is not clear on the material and methods. If they were incubated for 2 h, why were they incubated for less time than the samples from the previous experiment?

Point 8: Why were the irradiation doses different between the first samples and the samples from the LPS experiment? And they are also different for ALL patient samples and endometrial cancer patients.

Point 9: Why is RNA later added after incubation and not before?

Point 10: The work does not explain why a previous incubation of the samples is carried out before storage at -80ºC and RNA extraction? Why is RNA extraction not done directly?

Point 11: Regarding the housekeeping used, why has the HPTR1 gene been used and not a more common one such as beta actin or NADPH?

Point 12: Table S1 does not show the GeneBank accession numbers or the NDBI reference sequence from FDXR-207 to 220, where did you get the sequences to design the primers?

Point 13: In the statistical analysis, the authors who use paired t-test and one-way ANOVA, but have carried out previous analysis of normality and similarity of variances? Which? Was the data normal or not? Were the variances similar or not? Why do they carry out a logarithmic transformation? Explain in more detail. In the event that the data were not normal, performing the analysis with a non-parametric test would be more correct from the statistical point of view.

Point 14: In the statistical analysis they do not indicate that they have carried out a Pearson correlation analysis. However, Pearson correlation data appear in the results. Add to statistical analysis.

Point 15: - In the statistical analysis they do not indicate how they calculated the regression (Figure S5). However, it does appear in the results. Add this analysis to the statistical analysis.

Point 16: They indicate “Good linear correlation between dose and gene expression”. Remove "good" and add the value of the correlation.

Point 17: They indicate “best linear correlation for women…”. Delete "best".

Point 18: In Figure 1A, 2A and 2B, cannot be seen clearly. Increase the size of the figure or change the format.

Point 19: In Figure 1B, the Y axis is cut off. Authors must correct the figures.

Point 20: In figure 1 authors indicate that it has been carried out in 10 individuals, 5 women and 5 men. However, in the material and methods, section 2.5, they describe 9 “healthy donors”.

Point 21: The first lines of section 3.2 of the Results “We first generated….variants listed in Table S2” must be in material and methods.

Point 22: The experiments carried out should be in the same order to facilitate the understanding of the text within the material and methods and Results sections. For example, section 2.5 of material and methods “Blood irradiation ex vivo and RNA extraction for sequencing analysis” should be the last section of the results.

Point 23: In Figure 3, Control and LPS 500 ng / ml results are not clearly visible in the figure. Redo the figure.

Point 24: In Figure 4, authors should remove from the figure caption “One-way ANOVA followed by Turkey’s… in panel B”. The statistical analysis and the results must be described in the corresponding section.

Point 25: In Figure 5A, except for FDXR-201 +, the results cannot be clearly distinguished. Redo the figure so that they can be seen better.

Point 26: In section 3.3 of the results, they indicate variation coefficients, but in the statistical analysis they have not described this analysis. Add to statistical analysis. In addition, sometimes they indicate coefficients of variation in percentages and other times they do not. Unify the way in which the results are presented and analyze the coefficient of variation for all the variants.

Point 27: In section 3.4 of results, they use “3 healthy donors per replicate (total of 9 donors for 3 experiments”), but they are the same samples referred to in the material and methods. Which of them? What three experiments do the authors refer to? Why have some been chosen and not others? Define in material and methods which samples are used for sequencing analysis.

Point 28: In section of Discussion, "In vivo" and "Ex vivo" must always be in italics.

Point 29: In section of Discussion, the work has not analyzed alternative splicing in FDXR. Delete the phrase “and thoroughly investigated the complexities of alternative splicing in FDXR”.

Point 30: In section of Discussion, if there are no significant differences, delete the phrase “FDXR-206 was the most responsive to IR in men. FDXR-206 showed a similar trend in women, although there were no significant differences”. These results are not statistically proven, authors should not guess.

Point 31: Studies on "transcript variant response to IR exposure in cancer patients" have been previously published. I recommend that the authors read the review by Kai (2016) published in the International Journal of Molecular Science, for example.

Best regards,

Author Response

Reviewer 2

This work presents an original result with significant content about possible biomarkers of radiation exposure. However, the manuscript has some major changes that need to be corrected.

We thank the reviewer for his/her comments and suggestions. We have addressed all the comments and answered his/her questions below.

Specific comments:

Point 1: many parts of the text are highlighted in colour. Please remove these marks.

The parts highlighted in yellow marked all the changes made during the first revision. The highlighted areas now have been removed to avoid confusion.

Point 2: Please add line number to facilitate review.

We have incorporated the line numbers.

Point 3: The title "Introduction" should be in the same format as the rest of the titles of the work (bold type).

We have modified the format for the “introduction” title.

Point 4: Add a citation supporting the phrase "provide full-length transcripts which eliminates the issues of next-generation sequencing techniques related with mapping ambiguity."

We have added some citations to support our statement (Mantere et al., 2019 and Weirather et al., 2017)

Point 5: Authors should include information about the LPS effect in the introduction.

The following information has been added in the introduction about the confounding factor LPS:

Line 69 to 74

Some confounding factors have proven to affect radiation responsive genes such as simulated bacterial infection by lipopolysaccharide (LPS), curcumin, cancer condition or gender [8,9,19,20]. FDXR response to radiation has been observed to be temporarily counteracted by LPS [8] probably due to the downregulatory effect of LPS on P53 expression [21]. P53 is one of the main transcriptional factors regulating the DNA damage response and its target genes and potentially their variants could also be affected by LPS.

 Point 6: The sample size used in the study is low. Is there a bibliography about proven results with similar sample sizes? If not, the authors should include a larger number of samples.

The present study has used human blood samples for all the experiments and not cell lines. We believe that it provides more valuable data in terms of biomarker validation. For the dose-response curves, we have recruited 10 donors, 5 males and 5 females. Healthy donors come from volunteers who donate their blood at our centre (CRCE, Oxfordshire, UK) and we have limitations on the amount of blood and number of volunteers. We believe that 10 donors are a reasonable number for a study using human blood to be able to have conclusive results.

We also recruited 9 healthy donors to perform the sequencing experiments using Oxford nanopore technologies.  Data with 9 donors was already published earlier this year (Cruz-Garcia et al. 2020)

Regarding in vivo experiments, we provide data from 4 total body irradiated patients and 8 endometrial patients. These samples are very difficult to obtain and we believe that these numbers are adequate for this study.

To answer the reviewer comment, we listed below a non-exhaustive list of previous studies with similar sample sizes:

  1. Paul, S.; Amundson, S.A. Development of gene expression signatures for practical radiation biodosimetry. Int J Radiat Oncol Biol Phys 2008, 71, 1236-1244, doi:10.1016/j.ijrobp.2008.03.043.
  2. Budworth, H.; Snijders, A.M.; Marchetti, F.; Mannion, B.; Bhatnagar, S.; Kwoh, E.; Tan, Y.; Wang, S.X.; Blakely, W.F.; Coleman, M., et al. DNA repair and cell cycle biomarkers of radiation exposure and inflammation stress in human blood. PloS One 2012, 7, e48619, doi:10.1371/journal.pone.0048619.
  3. Cruz-Garcia, L.; O'Brien, G.; Sipos, B.; Mayes, S.; Love, M.I.; Turner, D.J.; Badie, C. Generation of a Transcriptional Radiation Exposure Signature in Human Blood Using Long-Read Nanopore Sequencing. Radiat Res 2020, 193, 143-154, doi:10.1667/rr15476.1.
  4. O’Brien, G.; Cruz-Garcia, L.; Majewski, M.; Grepl, J.; Abend, M.; Port, M.; Tichý, A.; Sirak, I.; Malkova, A.; Donovan, E., et al. FDXR is a biomarker of radiation exposure in vivo. Scientific Reports 2018, 8, 684, doi:10.1038/s41598-017-19043-w.
  5. Paul, S.; Barker, C.A.; Turner, H.C.; McLane, A.; Wolden, S.L.; Amundson, S.A. Prediction of in vivo radiation dose status in radiotherapy patients using ex vivo and in vivo gene expression signatures. Radiation research 2011, 175, 257-265, doi:10.1667/RR2420.1.
  6. Paul, S.; Smilenov, L.B.; Amundson, S.A. Widespread Decreased Expression of Immune Function Genes in Human Peripheral Blood Following Radiation Exposure. Radiation research 2013, 180, 575-583, doi:10.1667/RR13343.1.
  7. Cruz-Garcia, L.; O’Brien, G.; Donovan, E.; Gothard, L.; Boyle, S.; Laval, A.; Testard, I.; Ponge, L.; Wozniak, G.; Miszczyk, L., et al. Influence of Confounding Factors on Radiation Dose Estimation Using In Vivo Validated Transcriptional Biomarkers. Health Physics 2018, 115, 90-101, doi:10.1097/hp.0000000000000844.

Point 7: The blood samples from the LPS experiment were incubated for 2 or 24 hours? It is not clear on the material and methods. If they were incubated for 2 h, why were they incubated for less time than the samples from the previous experiment?

In line 97-98 we mention the incubation time of the LPS experiments: “Blood samples were kept at 37ºC in an incubator with 5% CO2 for 2 h and 24 h after exposure.”

In previous studies where we looked at the confounding effect of LPS on radiation responsive genes (O’Brien et al., 2018, Cruz-Garcia et al., 2018), we observed a temporary counteractive effect of LPS on FDXR upregulation after IR exposure 2h after LPS exposure. This effect is no longer detectable at 24h. This is the rationale to decide to keep the same time points to see whether FDXR variants would follow the same response.

O’Brien, G.; Cruz-Garcia, L.; Majewski, M.; Grepl, J.; Abend, M.; Port, M.; Tichý, A.; Sirak, I.; Malkova, A.; Donovan, E., et al. FDXR is a biomarker of radiation exposure in vivo. Scientific Reports 2018, 8, 684, doi:10.1038/s41598-017-19043-w.

Cruz-Garcia, L.; O’Brien, G.; Donovan, E.; Gothard, L.; Boyle, S.; Laval, A.; Testard, I.; Ponge, L.; Wozniak, G.; Miszczyk, L., et al. Influence of Confounding Factors on Radiation Dose Estimation Using In Vivo Validated Transcriptional Biomarkers. Health Physics 2018, 115, 90-101, doi:10.1097/hp.0000000000000844.

Point 8: Why were the irradiation doses different between the first samples and the samples from the LPS experiment? And they are also different for ALL patient samples and endometrial cancer patients.

All the ex vivo experiments received a single 2 Gy X-ray dose (at a dose rate of 0.5 Gy/min). The IR dose of 2 Gy was chosen due to its wide use as therapeutic dose per fraction delivered during radiotherapy treatments; besides this dose is largely used for triage purposes to identify individuals at risk of developing acute radiation syndrome. Besides, we performed dose-response experiments where the samples were exposed to lower or higher doses than 2 Gy (0.25, 0.5, 1, 2, 3 and 4 Gy).

Regarding the in vivo samples, the endometrial patients received 1.8 Gy fractions and the TBI patients between 1.6 and 2 Gy fractions. All close to the 2 Gy dose.

Hence the dose of 2 Gy is relevant as individuals exposed to this level of dose would require rapid medical attention.

For the reasons mentioned above we believe that this is the most relevant dose we could choose. This is also a dose used by other research groups worldwide for blood-based gene expression studies; there are a large amount of existing data to compare with on already validated transcriptional biomarkers of radiation exposure.

Point 9: Why is RNA later added after incubation and not before?

The purpose of RNA later is to stabilize and protect cellular RNA from the samples before storing them. RNA later denatures proteases and RNases to preserve RNA. It can’t be used before incubation.

Point 10: The work does not explain why a previous incubation of the samples is carried out before storage at -80ºC and RNA extraction? Why is RNA extraction not done directly?

We understand that ex vivo and in vivo samples which follow different protocols might be confusing; we try to provide a clear explanation below:

For ex vivo experiments with IR exposure, samples were incubated for 2h or 24h after exposure to allow the cells to activate the DNA damage response which  triggers the ATM\CHECK2\P53 transcriptional pathway activation; P53 regulates transcriptionally the gene FDXR after exposure. It is a common protocol followed in this type of studies, some examples:

  1. Paul, S.; Amundson, S.A. Development of gene expression signatures for practical radiation biodosimetry. Int J Radiat Oncol Biol Phys 2008, 71, 1236-1244, doi:10.1016/j.ijrobp.2008.03.043.
  2. Budworth, H.; Snijders, A.M.; Marchetti, F.; Mannion, B.; Bhatnagar, S.; Kwoh, E.; Tan, Y.; Wang, S.X.; Blakely, W.F.; Coleman, M., et al. DNA repair and cell cycle biomarkers of radiation exposure and inflammation stress in human blood. PloS One 2012, 7, e48619, doi:10.1371/journal.pone.0048619.
  3. Cruz-Garcia, L.; O'Brien, G.; Sipos, B.; Mayes, S.; Love, M.I.; Turner, D.J.; Badie, C. Generation of a Transcriptional Radiation Exposure Signature in Human Blood Using Long-Read Nanopore Sequencing. Radiat Res 2020, 193, 143-154, doi:10.1667/rr15476.1.
  4. O’Brien, G.; Cruz-Garcia, L.; Majewski, M.; Grepl, J.; Abend, M.; Port, M.; Tichý, A.; Sirak, I.; Malkova, A.; Donovan, E., et al. FDXR is a biomarker of radiation exposure in vivo. Scientific Reports 2018, 8, 684, doi:10.1038/s41598-017-19043-w.
  5. Paul, S.; Barker, C.A.; Turner, H.C.; McLane, A.; Wolden, S.L.; Amundson, S.A. Prediction of in vivo radiation dose status in radiotherapy patients using ex vivo and in vivo gene expression signatures. Radiation research 2011, 175, 257-265, doi:10.1667/RR2420.1.
  6. Paul, S.; Smilenov, L.B.; Amundson, S.A. Widespread Decreased Expression of Immune Function Genes in Human Peripheral Blood Following Radiation Exposure. Radiation research 2013, 180, 575-583, doi:10.1667/RR13343.1.
  7. Cruz-Garcia, L.; O’Brien, G.; Donovan, E.; Gothard, L.; Boyle, S.; Laval, A.; Testard, I.; Ponge, L.; Wozniak, G.; Miszczyk, L., et al. Influence of Confounding Factors on Radiation Dose Estimation Using In Vivo Validated Transcriptional Biomarkers. Health Physics 2018, 115, 90-101, doi:10.1097/hp.0000000000000844.

Once the incubation period is finished, we don’t process samples straight away, we stored them in RNA later at -80C. After collecting samples, performing the experiments and the incubation periods, it is not always possible to extract the samples straight away because this process would continue during unsocial hours after already long days of experiments. We also don’t have staff working overnights in our institution.

For in vivo samples, blood samples are collected in PAXGene tubes directly where lysis takes place. For these samples the incubation occurs ‘in vivo’.

Point 11: Regarding the housekeeping used, why has the HPTR1 gene been used and not a more common one such as beta actin or NADPH?

HPRT1 is used in studies using blood and irradiation because it is the most stable gene under these conditions. Some references about the use of HPRT1 for this type of experiments:

  1. Manning, G.; Macaeva, E.; Majewski, M.; Kriehuber, R.; Brzóska, K.; Abend, M.; Doucha-Senf, S.; Oskamp, D.; Strunz, S.; Quintens, R., et al. Comparable dose estimates of blinded whole blood samples are obtained independently of culture conditions and analytical approaches. Second RENEB gene expression study. International Journal of Radiation Biology 2017a, 93, 87-98, doi:10.1080/09553002.2016.1227105.
  2. Abend, M.; Badie, C.; Quintens, R.; Kriehuber, R.; Manning, G.; Macaeva, E.; Njima, M.; Oskamp, D.; Strunz, S.; Moertl, S., et al. Examining Radiation-Induced In Vivo and In Vitro Gene Expression Changes of the Peripheral Blood in Different Laboratories for Biodosimetry Purposes: First RENEB Gene Expression Study. Radiation Research 2016, 185, 109-123, doi:10.1667/RR14221.1.
  3. Kabacik, S.; Manning, G.; Raffy, C.; Bouffler, S.; Badie, C. Time, Dose and Ataxia Telangiectasia Mutated (ATM) Status Dependency of Coding and Noncoding RNA Expression after Ionizing Radiation Exposure. Radiation Research 2015a, 183, 325-337, doi:10.1667/RR13876.1.
  4. Tichy, A.; Kabacik, S.; O’Brien, G.; Pejchal, J.; Sinkorova, Z.; Kmochova, A.; Sirak, I.; Malkova, A.; Beltran, C.G.; Gonzalez, J.R., et al. The first in vivo multiparametric comparison of different radiation exposure biomarkers in human blood. PLOS ONE 2018, 13, e0193412, doi:10.1371/journal.pone.0193412.
  5. Manning, G.; Kabacik, S.; Finnon, P.; Bouffler, S.; Badie, C. High and low dose responses of transcriptional biomarkers in ex vivo X-irradiated human blood. International Journal of Radiation Biology 2013, 89, 512-522, doi:10.3109/09553002.2013.769694.
  6. Kabacik, S.; Mackay, A.; Tamber, N.; Manning, G.; Finnon, P.; Paillier, F.; Ashworth, A.; Bouffler, S.; Badie, C. Gene expression following ionising radiation: identification of biomarkers for dose estimation and prediction of individual response. International Journal of Radiation Biology 2011a, 87, 115-129, doi:10.3109/09553002.2010.519424.
  7. Kabacik, S.; Ortega-Molina, A.; Efeyan, A.; Finnon, P.; Bouffler, S.; Serrano, M.; Badie, C. A minimally invasive assay for individual assessment of the ATM/CHEK2/p53 pathway activity. Cell Cycle (Georgetown, Tex.) 2011b, 10, 1152-1161, doi:10.4161/cc.10.7.15231.
  8. Manning, G.; Tichý, A.; Sirák, I.; Badie, C. Radiotherapy-Associated Long-term Modification of Expression of the Inflammatory Biomarker Genes ARG1, BCL2L1, and MYC. Frontiers in Immunology 2017b, 8, 412, doi:10.3389/fimmu.2017.00412.

As one of the rules we follow for quantitative PCR experiments, we use a housekeeping gene which level of endogenous expression is as close as possible to the basal level of the genes of interested and Cts of PCR similar for better analyses. Therefore, HPRT was chosen. The transcriptional expression level of Beta Actin or NAPDH is much higher in white blood cells and not adequate.

Point 12: Table S1 does not show the GeneBank accession numbers or the NDBI reference sequence from FDXR-207 to 220, where did you get the sequences to design the primers?

The sequences from FDXR-207 to 220 are only available in Ensembl, and the transcript IDs are available in Table S1. All the sequences are available following the transcript ID in Ensembl genome browser.

Point 13: In the statistical analysis, the authors who use paired t-test and one-way ANOVA, but have carried out previous analysis of normality and similarity of variances? Which? Was the data normal or not? Were the variances similar or not? Why do they carry out a logarithmic transformation? Explain in more detail. In the event that the data were not normal, performing the analysis with a non-parametric test would be more correct from the statistical point of view.

Thanks to the reviewer for this comment; Classically we always use log transformed data to bring gene expression values to a normal distribution. Log transformation permits to obtain more symmetrical data and therefore, a parametric statistical test is adequate and  provides a more accurate and relevant analysis.

Point 14: In the statistical analysis they do not indicate that they have carried out a Pearson correlation analysis. However, Pearson correlation data appear in the results. Add to statistical analysis.

We apologised for this omission and more information has now been added in the statistical analysis section:

“Pearson's correlation and linear regression analyses were performed to verify dose-response relationships.” Line 179-180

Point 15: - In the statistical analysis they do not indicate how they calculated the regression (Figure S5). However, it does appear in the results. Add this analysis to the statistical analysis.

Again, thanks for this and more information has been added in the statistical analysis section:

“Pearson's correlation and linear regression analyses were performed to verify dose-response relationships.” Line 179-180

Point 16: They indicate “Good linear correlation between dose and gene expression”. Remove "good" and add the value of the correlation.

Line 205: “good” has now been removed from the sentence.

Point 17: They indicate “best linear correlation for women…”. Delete "best".

Line 207: the sentence has been modified and “best” being removed: The variant presenting the linear correlation with higher R vale and lower p-vale for women was FDXR-217

Point 18: In Figure 1A, 2A and 2B, cannot be seen clearly. Increase the size of the figure or change the format.

We agree with the reviewer and all the Tukey pairwise comparisons letters have been removed from the figures and are now available in tables in supplementary material (Table S3 and Table S4).

Point 19: In Figure 1B, the Y axis is cut off. Authors must correct the figures.

This has been done on purpose. The Y axis in the figure 1 B has a scale break to be able to ‘zoom in’ in order to better see the low expressed variants. Scale breaks are used to display two distinct ranges in the same chart area.

Point 20: In figure 1 authors indicate that it has been carried out in 10 individuals, 5 women and 5 men. However, in the material and methods, section 2.5, they describe 9 “healthy donors”.

Section 2.5 refers to Nanopore sequencing analyses which have been performed with 9 donors. Figure 1 has been produced with 10 independent donors corresponding to section 2.1.

Point 21: The first lines of section 3.2 of the Results “We first generated….variants listed in Table S2” must be in material and methods.

We agree, and this paragraph has been moved to material and method section 2.4.

Point 22: The experiments carried out should be in the same order to facilitate the understanding of the text within the material and methods and Results sections. For example, section 2.5 of material and methods “Blood irradiation ex vivo and RNA extraction for sequencing analysis” should be the last section of the results.

Material and methods and results sections follow the exact same order. The example mentioned for instance, section 2.5 in material in methods describes the last section in the results 3.4.

Point 23: In Figure 3, Control and LPS 500 ng / ml results are not clearly visible in the figure. Redo the figure.

We believe that the results in Figure 3 are clearly visible,; two-time points, 2h and 24h, are presented and each graph is showing one variant. It would not be possible to see each individual value and the statistical differences. We believe that modifying this particular figure would not bring a significant improvement.

Point 24: In Figure 4, authors should remove from the figure caption “One-way ANOVA followed by Turkey’s… in panel B”. The statistical analysis and the results must be described in the corresponding section.

The figure legends need to be as much informative as possible and they require to describe the statistical analysis performed.

Point 25: In Figure 5A, except for FDXR-201 +, the results cannot be clearly distinguished. Redo the figure so that they can be seen better.

We believe that dot plots are a classic way to present this type of data. They are the most honest, transparent and unambiguous way to present these data, following examples similar data already reported. The consistency of the results in figure 5A is reflected by the fact that the dots group together and this is an information that would be more difficult to clearly visualise with other representations like bar charts. We therefore believe, that the figure is well presented and doesn’t require modification.

Point 26: In section 3.3 of the results, they indicate variation coefficients, but in the statistical analysis they have not described this analysis. Add to statistical analysis. In addition, sometimes they indicate coefficients of variation in percentages and other times they do not. Unify the way in which the results are presented and analyze the coefficient of variation for all the variants.

We agree with the reviewer and the % in line 294 has been removed because CV has no units. We have unified the results presentation as requested and included the CV information in the statistical section:

Line 187-190: To measure the consistency of the gene expression response to radiation in the in vivo experiments and sequencing analysis, the coefficient of variation (CV, mean-normalized standard deviation) was calculated for all the variants.

Point 27: In section 3.4 of results, they use “3 healthy donors per replicate (total of 9 donors for 3 experiments”), but they are the same samples referred to in the material and methods. Which of them? What three experiments do the authors refer to? Why have some been chosen and not others? Define in material and methods which samples are used for sequencing analysis.

We are clarifying the sample information: Three independent experiments were performed including 3 donors in each experiment. To be able to perform the experiments with polyA+ RNA which was necessary to generate enough data, we had to use 3 donors per biological replicate and this protocol has been previously published (Cruz-Garcia et al., 2020). Poly+ RNA constitutes only a very small proportion of the total RNA, around 1-5% of total RNA. The section 2.5 in material and methods describe specifically the results in section 3.4. The other experiments and donors are not related to this specific sequencing experiment. Section 2.5 title states that the section describes specifically the sequencing analysis: “Blood irradiation ex vivo and RNA extraction for sequencing analysis”.

Cruz-Garcia, L.; O'Brien, G.; Sipos, B.; Mayes, S.; Love, M.I.; Turner, D.J.; Badie, C. Generation of a Transcriptional Radiation Exposure Signature in Human Blood Using Long-Read Nanopore Sequencing. Radiat Res 2020, 193, 143-154, doi:10.1667/rr15476.1.

Point 28: In section of Discussion, "In vivo" and "Ex vivo" must always be in italics.

All the "In vivo" and "Ex vivo" have been reviewed in the manuscript and corrected as requested.

Point 29: In section of Discussion, the work has not analyzed alternative splicing in FDXR. Delete the phrase “and thoroughly investigated the complexities of alternative splicing in FDXR”.

Line 332 has been modified: and provided the first comprehensive analysis of the transcriptional complexities of the FDXR variants’

Point 30: In section of Discussion, if there are no significant differences, delete the phrase “FDXR-206 was the most responsive to IR in men. FDXR-206 showed a similar trend in women, although there were no significant differences”. These results are not statistically proven, authors should not guess.

The following sentence has been removed: FDXR-206 showed a similar trend in women, although there were no significant differences.

Point 31: Studies on "transcript variant response to IR exposure in cancer patients" have been previously published. I recommend that the authors read the review by Kai (2016) published in the International Journal of Molecular Science, for example.

We believe that we have accessed the manuscript suggested by the reviewer with the information he provided and found the publication: “Role of RNA-Binding proteins in DNA damage response” (M.Kai, 2016, IJMS). To our understanding, the manuscript reviews how FET proteins are recruited in the DNA damage site and play an important role in PARP-dependent DNA damage response process. Therefore, this manuscript Kai 2016 doesn’t refer to “transcript variant response to IR exposure in cancer patients”.  Perhaps the reviewer could provide the full reference of the publication reporting transcript variants in response to ionising radiation in cancer patients specifically.

Kai, M. Roles of RNA-Binding Proteins in DNA Damage Response. Int J Mol Sci 2016, 17, 310, doi:10.3390/ijms17030310.

Best regards,

Round 2

Reviewer 2 Report

Thank you very much for the changes made and for the answers given to my questions.

This manuscript is a resubmission of an earlier submission. The following is a list of the peer review reports and author responses from that submission.

Round 1

Reviewer 1 Report

The presented study touches on one of the most unresearched areas of genomics: the functional significance and regulation of alternative variants of gene splicing. Using variant-specific primers for RT-PCR and new approach with nanopore sequencing, the authors evaluate the background and radioinduced expression levels of the splice variants of the FDXR gene. Despite the lack of attempts to elucidate the functional differences between the variants, the presented work is a small step towards understanding the poorly studied field of genomics.

Minor points:

1) Abstract: I think that the FDXR gene name should be decrypted in an abstract so that the abstract is more easily understood separately from the main text.
2) Have you analyzed for potential differences in background and induced expression levels of different splice variants of FDXR between samples received from womens and mens? Are there any significant differences? Is it right to combine this data into one selection?
3) Figure 1: Figures B-I duplicate the information presented in Figure A. I think the authors should save only Figure 1A and combine it with Figure 2. Moreover, if in Figure 2 authors will use the same color code as in Figure 1A, then the perception of the results will be much easier.
4) The organisation of figure 5 is confusing. I think that all splice variants can be simply divided into two graphs: highly expressed and low expressed, without repetition.
5) One of the splice variants presented in the NCBI (NM_001258016.3) has an alternative transcription start site. In my opinion, the authors should discuss his reaction to radiation in comparison with other variants, that have the same TSS. To a small extent, this will make it possible to evaluate the role in the regulation of expression not only of splicing itself, but also of differences in the promoter.

Reviewer 2 Report

The authors analyzed human blood to characterize the ionizing radiation response of the alternative transcripts of a single gene, FDXR, a gene previously known to be responsive under various radiation conditions.

They have performed ex vivo radiation of blood and found convincing dose response curves for a number of variants that had quite varied basal levels. The resulting comments in the results about fold change would have been bolstered by some p-values although the curves in themselves are noted to be statistically sound. They also have looked at blood samples from patients that have had total body irradiation (n=4) and patients who had targeted radiotherapy (n=8). It is clear from the error bars that the patients treatment conditions vary. Although comment on increases are common, again no p-values have been reported between samples and the experimental samples have large error bars. It was interesting that a couple variants showed some consistency in targeted radiotherapy, but statistical comparison to controls was not comprehensive. They have also conducted some variant analysis with sequencing and found additional low expressed variants. Not all statistical methods are provided in the Methods section. No additional experimentation or functional tests beyond transcriptional analysis in response to IR has been completed.

Some figures such as the sequencing data in Fig 5 is not well presented. A rethink in presenting this information would be useful. Also, the figure legends are often missing information. The legends need to be complete and stand on their own. They conclude that they characterized the transcript variants of FDXR in response to radiation in vivo. They also comment that the role of the variants is unknown but the responses could be used as biomarkers for radiation exposure. These aspects were not tested in this body of work. The area looking into the function of the variants that are selectively expressed after radiation treatment is interesting, and the data provided in this paper provides a direction towards future experimentation.

The Discussion section is very weak and the authors have not given a comprehensive discussion of the results. The full meaning, implications and concerns are just not covered well enough. The conclusions stated are weak as well. There is a need to minimise results and methods that have crept into the Discussion section, and be sure to include all methods in the Methods section. Re-writing is required in most sections.

Some addressment on the regulatory/functional regions in the protein / spliced product that could provide an explanation for the variant increase or reason for response to radiation would benefit the paper. Also, what concerns were there about the variation found between the patients? How does this affect the results. Some discussion on the low expressed variants that were found with the sequencing needs some direct comments and discussion. Some reasonable discussion of the results (meaning) that lead to the next step to understand these variants or make use of them for future methods in radiotherapy etc. and how this relates to other studies including comparison to other genes in more depth. Etc.

Why are LPS results presented in the supplementary section and not as a figure in the paper since a section in the results is given to this aspect?